# Crash Analysis of Aluminum/CFRP Hybrid Adhesive Joint Parts Using Adhesive Modeling Technique Based on the Fracture Mechanics

**DOI:** 10.3390/polym13193364

**Published:** 2021-09-30

**Authors:** Young Cheol Kim, Soon Ho Yoon, Geunsu Joo, Hong-Kyu Jang, Ji-Hoon Kim, Mungyu Jeong, Ji Hoon Kim

**Affiliations:** 1Department of Composite Structure & System, Korea Institute of Materials Science, KIMS, 797, Changwon-daero, Seongsan-gu, Changwon-si 51508, Korea; yckim@kims.re.kr (Y.C.K.); gsjoo@kims.re.kr (G.J.); hongkyu@kims.re.kr (H.-K.J.); jhkim01@kims.re.kr (J.-H.K.); mgjeong@kims.re.kr (M.J.); 2Department of Mechanical Engineering, Pusan National University, Pusan 46241, Korea; kimjh@pusan.ac.kr

**Keywords:** structural adhesive, carbon-fiber-reinforced plastic (CFRP), fracture toughness, crash simulation, cohesive zone model, tiebreak contact

## Abstract

This study describes the numerical simulation results of aluminum/carbon-fiber-reinforced plastic (CFRP) hybrid joint parts using the explicit finite-element solver LS-DYNA, with a focus on capturing the failure behavior of composite laminates as well as the adhesive capacity of the aluminum–composite interface. In this study, two types of adhesive modeling techniques were investigated: a tiebreak contact condition and a cohesive zone model. Adhesive modeling techniques have been adopted as a widely commercialized model of structural adhesives to simulate adhesive failure based on fracture mechanics. CFRP was studied with numerical simulations utilizing LS-DYNA MAT54 to analyze the crash capability of aluminum/CFRP. To evaluate the simulation model, the results were compared with the force–displacement curve from numerical analysis and experimental results. A parametric study was conducted to evaluate the effect of different fracture toughness values used by designers to predict crash capability and adhesive failure of aluminum/CFRP parts.

## 1. Introduction

Automotive structural parts are being replaced by lightweight materials, such as carbon-fiber-reinforced plastic (CFRP), plastics, and aluminum, instead of steel, to improve fuel efficiency and reduce carbon emissions in the automotive industry [1,2]. Due to the fact that mechanical joining methods, such as bolts or welding, are unsuitable for these materials, a structural adhesive is a good alternative to provide the required strength at joints for dissimilar materials [3]. In addition, it is important to predict the performance of the adhesive for joints by applying joining technology between different materials to automotive structures [4].

Many studies have been conducted to predict and evaluate the strength of adhesive joints using continuum mechanics and fracture mechanics approaches [5]. The continuum mechanics approach has been used to analyze the strength of the adhesive; it requires stress distributions and adequate failure criteria [6]. However, it is difficult to apply the stress or strain to the design of the structure because the stress or strain cannot be defined analytically, precisely owing to the stress singularity of the adhesive joint [7]. The cohesive zone model is based on fracture mechanics and it assumes that there is a softening zone in front of the crack tip. In the fracture process zone, the crack tip opening is resisted by tractions. Early conceptual work was conducted by Dugdale [8] and Barenblatt [9]. Hillerborg et al. [10] applied the cohesive zone formulation to cracking in a concrete beam.

Over the years, many adhesive modeling methods have been developed for adhesively bonded joints used in crash simulation [11,12]. Faruque et al. [13] proposed a practical modeling methodology for adhesively bonded structures using discrete springs for crash simulation. Dlugosch et al. [14] tested hybrid FRP (fiber-reinforced plastic)–steel tubes under dynamic axial loading and conducted the numerical analyses using Abaqus Explicit. To study their predictability, the adhesives used between the steel and the FRP interface were modeled using cohesive behavior and tied surfaces modeling methods. Shin et al. [15] investigated the damage behavior of an aluminum/composite beam under bending conditions by conducting a finite-element analysis. Debonding and delamination were modeled by a cohesive zone model. May et al. [16] proposed a rate-dependent constitutive cohesive law for the model. The model was validated as a test of the T-joint with high-strength steel and structural adhesive under quasi-static and dynamic loading conditions.

In this study, explicit dynamic analysis software, LS-DYNA, was used to analyze the joint performance of the aluminum/CFRP parts by using two adhesive modeling techniques based on fracture mechanics (the cohesive zone model and tiebreak contact condition). The results of crash tests and finite-element analysis were compared and analyzed. To define the material model, a fracture toughness test of the adhesive was performed. The results were then used to evaluate the strength analysis of parts under impact conditions, the failure of the composite material, and the failure behavior of the adhesive. The validity of the analysis model was verified. The purpose of this study was to develop a practical method to model a large-scale, adhesively bonded joint structure with a simple procedure and acceptable computational costs based on the existing modeling approaches.

## 2. Mechanical Properties of Aluminum, CFRP Plates, and Structural Adhesive

### 2.1. Mechanical Properties of Aluminum, CFRP Plates

Aluminum 5052-O has good formability and ductility, and it was used to increase the impact absorption capability [17]. The material properties of aluminum 5052-O (Korea Non-Ferrous Metals Corporation, Asan, Korea) are presented in Table 1.

The CFRP plates (SHINSUNG BASIC MATERIALS, Anseong, Korea) were made from eight plies of CFRP with a stacking sequence of [0]_8_. The plates were manufactured using a pultrusion manufacturing process [21]. The tensile, compression, and shear stiffness and strength tests were performed according to ASTM D3039, ASTM D6641, and ASTM D7078 standards to secure the material properties [22,23,24]. A material testing machine, Instron model 5985 (Instron Corporation, Norwood, MA, USA), was used to run tests. The obtained test results are presented in Table 2.

### 2.2. Fracture Toughness of Structural Adhesive

The fracture toughness test of the adhesive was performed to apply the critical energy release rate to finite-element analysis. The fracture toughness Mode I test was performed according to the ASTM D3433 standard [25]. In the case of the Mode II test, the fracture toughness value was measured using the tapered end-notched flexure (TENF) test method [26]. Figure 1 shows the dimensions of the specimen and fracture toughness test setup. In these tests, a urethane-based vehicle structural adhesive developed by Dongsung Chemical was used, and high-strength steel (STD-11, SeAH css Corporation, Changwon, Korea) material was used for the adherend. Additionally, the testing machine, Instron model 5882, was used to conduct fracture toughness tests. As a result of the fracture toughness tests, the values of 2.010 kJ/m^2^ for Mode I and 7.666 kJ/m^2^ for Mode II were obtained. The test results are presented in Table 3.

## 3. Aluminum/CFRP Component Test

Hat-profile specimens were fabricated to carry out crash tests on aluminum/CFRP hybrid joint parts. The Al5052-O aluminum alloy material (Korea Non-Ferrous Metals Corporation, Asan, Korea) with a thickness of 2.5 mm was manufactured by applying a bending manufacturing process. CFRP material of 2.0 mm was bonded using a structural adhesive [27]. The dimensions of the aluminum/CFRP hybrid joint parts are shown in Figure 2 and Table 4.

The crash test was performed by dropping a semicircular impactor with a weight of 47.1 kg from a height of 2.3 m to impose an impact at an initial speed of 6.38 m/s. The crash test setup is shown in Figure 3. The speed and displacement were measured using a photonic sensor and a rotary encoder sensor. An aluminum/CFRP specimen was installed on the supporting parts made of a hardened steel tool.

Figure 4 shows the failure of the aluminum/CFRP structure after the crash test. The aluminum parts had large plastic deformations that occurred while absorbing energy after a crash. The CFRP was damaged as a result of the excessive deformation of the aluminum part while supporting the impact load. Bending failure occurred in fiber and transverse directions. Tearing failure was observed at the corners of the aluminum, which was caused by a reduction in the width of the aluminum plate material due to bending during the manufacturing process [28].

In the graphs of the crash test results in Figure 5, the load time interval slightly decreased from t = 0.005 s to t = 0.010 due to the failure of the CFRP as well as the failure of the adhesive between the aluminum and the CFRP.

## 4. Finite-Element Analysis and Verification

### 4.1. Material Models and Finite-Element Model

LS-DYNA (Ansys Inc, Canonsburg, PA, USA, which is explicit dynamic analysis software, was used to create a finite-element model of the aluminum/CFRP component crash test. The finite-element model is shown in Figure 6 in the same manner as the test conditions for crash analysis.

The material model MAT20 (MAT_RIGID) was used to model the impactor in LS-DYNA. The impactor was constrained in the X, Y displacement, excluding the Z direction, which was the impact load direction, with an initial speed of 6.38 m/s, corresponding to 959 J. An automatic single surface contact option was used to prevent interpenetration for the contact condition. LS-DYNA provides various anisotropic material models related to the composites [29]. In this study, the MAT54 (*MAT_ENHANCED_COMPOSITE_DAMAGE) material card was used to model the CFRP. The MAT54 material model is widely used in the industrial field and is effective because it has simple input parameters and damage models for the failure mechanisms of complex composite materials, as shown in Figure 7. Elastic modulus (EA, EB, EC), Poisson’s ratio (PRBA, PRCA, PRCB), and shear modulus (GAB, GBC, GCA) indicate elastic material properties (yellow section) in input parameters. The notations of A, B, and C indicate material direction. In addition, the input parameters of strength (blue section) are designated for each direction. XC and XT define the compressive and tensile strengths for the fiber direction. YC and YT denote compressive and tensile strengths for the matrix direction. The shear strength can be introduced by a parameter of SC. Experimental tensile tests, compressive tests, and shear tests are used to determine the mechanical properties of composite materials. Input parameters of elastic and the strength of material properties are not involved in the calibration of input parameters [30].

Chang–Chang criteria are two-dimensional failure criteria. It has been proposed to predict the progressive damage of composite structures under loading [31]. The strength parameters were applied to define the onset of ply degradation using the Chang–Chang failure criterion in MAT54. The Chang–Chang failure criteria for composite materials adhere to the following conditions [32]:

Tensile fiber failure mode:


(1)
$$\sigma_{aa}>0\;then\;e_{f}^{2}={\left(\frac{\sigma_{aa}}{X_{t}}\right)}^{2}+\beta {\left(\frac{\sigma_{ab}}{S_{c}}\right)}^{2}-1,\;\left\{\frac{e_{f}^{2}\;\geq 0\;:failed}{e_{f}^{2}<0\;:elastic}\right.\;after\;failure\;E_{a}=E_{b}=G_{ba}=v_{ab}=v_{ba}=0;$$


Compressive fiber failure mode:


(2)
$$\sigma_{aa}<0\;then\;e_{c}^{2}={\left(\frac{\sigma_{aa}}{X_{t}}\right)}^{2}-1,\;\left\{\frac{e_{f}^{2}\;\geq 0\;:failed}{e_{f}^{2}<0\;:elastic}\right.\;after\;failure\;E_{a}=v_{ab}=v_{ba}=0;$$


Tensile matrix failure mode:


(3)
$$\sigma_{bb}>0\;then\;e_{m}^{2}={\left(\frac{\sigma_{bb}}{Y_{t}}\right)}^{2}+{\left(\frac{\sigma_{ab}}{S_{c}}\right)}^{2}-1,\;\left\{\frac{e_{f}^{2}\;\geq 0\;:failed}{e_{f}^{2}<0\;:elastic}\right.\;after\;failure\;E_{b}=v_{ba}=0\;\Rightarrow G_{ba}=0;$$


Compressive matrix failure mode:


(4)
$$\sigma_{bb}<0\;then\;e_{d}^{2}={\left(\frac{\sigma_{bb}}{2S_{c}}\right)}^{2}+\left[{\left(\frac{Y_{c}}{2S_{c}}\right)}^{2}-1\right]\frac{\sigma_{bb}}{Y_{c}}+{\left(\frac{\sigma_{ab}}{S_{c}}\right)}^{2}-1,\;\left\{\frac{e_{f}^{2}\;\geq 0\;:failed}{e_{f}^{2}<0\;:elastic}\right.\;after\;failure\;E_{b}=v_{ab}=v_{ba}=0\Rightarrow G_{ba}=0;$$


In the MAT54 composite material model, failure criteria are related to failure strain parameters in the tensile/compression direction of fibers and matrix, such as DFAILT, DFAILC, DFAILM, DFAILS, and effective failure strain (EFS). To apply the simple failure criterion of the composite material in complex deformation behavior, the EFS value, which is the overall failure strain criterion, was applied as a value of 0.3 through a trial-and-error method, and the TFAIL and SOFT values, which are nonphysical parameters, were set to zero [33]. In the case of the aluminum material model, MAT24 (*MAT_PIECEWISE_LINEAR_PLASTICITY) was set, which is a material model that generally reflects the characteristics of the elastic–plastic behavior well. It can also define the failure criterion according to the stress–strain relationship [34,35].

### 4.2. Cohesive Zone Model

The fracture process zone is modeled as a cohesive zone [36]. The fracture characteristics are defined by the traction–separation law, constituting the cohesive element. The dissipated energy of the traction–separation relationship is equal to the critical energy release rate, which is the energy required for crack propagation [37].

In Figure 8, a few cohesive zone material models are presented that can be used to model adhesive bonds in LS-DYNA. The solid elements of the ELFORM 20 are intended for use in cohesive material models. Depending on the situation, the cohesive zone model can be applied to modeling to determine the plastic properties of the adhesive and the rate-dependency properties. In this study, the adhesive was modeled using MAT138 (*MAT_COHESIVE_MIXED _MODE), a cohesive zone model defined by the bilinear traction–separation relationship [38]. The cohesive zone model of the aluminum/CFRP hybrid joint parts is shown in Figure 9.

### 4.3. Tiebreak Contact

Adhesive debonding can also be modeled using the tiebreak contact condition between the adherends in LS-DYNA. Tiebreak contact is a penalty-based contact condition modeling technique. It is useful when constraints are applied to parts with different meshes and exists within the master segment projection area based on the slave node to define the contact condition. The interval between the slave node and the master segment is a specific value based on the dimension of the element. A contact option of CONTACT_AUTOMATIC_ ONE_WAY_SURFACE_TO_SURFACE_TIEBREAK was used for the finite-element model. The failure criteria were the same as those of MAT138. After the tiebreaking process, it becomes an automatic contact condition [39]. In the cohesive zone model, it is inconvenient to connect the nodes by modeling the adhesive in the joint area as a solid; however, it is easy to model the adhesive using the tiebreak contact condition. This is possible with simple conditions, such as designating a segment area, or parts on the surface of the joint [40].

### 4.4. Finite-Element Analysis Results and Verification

Figure 10 shows the failure of the aluminum/CFRP parts, sequentially. It was confirmed that the main energy is absorbed by plastic deformation on the impacted region of the aluminum/CFRP part. The large deformation caused the edge part to tear and break the composite material and adhesive simultaneously. In the graph in Figure 11, a section exists where the load decreased as the adhesive at the aluminum–composite interface failed. Although the section where the composite material failure is different in the cohesive zone model and the tiebreak contact condition, the trend of the impact load is similar. In Figure 12 and Figure 13, the photographs show the composite failure and finite-element analysis results. The fracture of the composite material occurred in the fiber direction and the transverse of the fiber. In the case of adhesive failure (Figure 14), the cohesive zone model had a wider debonding area than in the tiebreak contact condition. Energy absorption parameters for a structure to evaluate its performance under crash loading require the definition of some indicators. Generally, this parameter can be determined from the load–displacement curve. It is the area of the force–displacement curve in a crash situation and can be calculated as:(5)∫F dδ
where F is the crushing force and δ is the crushing depth. In addition, the crushing displacement of the impactor can be characteristic of the energy absorption capability at the same amount of impact energy. Table 5 shows the energy absorption and the crush depth of the impactor.

### 4.5. Effect of Mesh Size on Finite-Element Analysis of Adhesive Joint

The mesh size of the finite-element analysis is one of the most significant limitations of the cohesive zone model method. It has been observed that it is essential to include between two and three interface elements in the cohesive zone model to precisely represent the softening ahead of the fracture process zone [41]. Analyses that violate this condition show a characteristic stick–slip behavior after failures. This violation results in an incorrect and uncertain solution [42,43]. To create a reliable, finite model with the appropriate mesh size, it is necessary to investigate the mesh dependency of a model. Figure 15 and Table 6 show the results of finite-element analysis for different mesh sizes. The results in the case of mesh size 2 mm are acceptable based on computational costs and prediction accuracy. In the case of a mesh size of 5 mm, since no failure of composite material occurred, it was not possible to observe a structure with a decreased structural rigidity in the graph.

### 4.6. Effect of Fracture Toughness on Finite-Element Analysis of Adhesive Joint

A parametric study was conducted to assess the effect of different fracture toughness values used by engineers to predict the impact strength and adhesive failure of aluminum/CFRP components. Parameters of fracture toughness values for the material models of the adhesive in the simulation were used to investigate the impact strength and adhesive failure of the aluminum/CFRP hybrid adhesive joint parts according to the adhesive fracture toughness values. Case I had low fracture toughness values, and Case II had high fracture toughness values. They were divided into two cases and analyzed for comparison. In general, because the fracture toughness value of Mode II was approximately three to four times that of Mode I [44], the fracture toughness values for each case were set as shown in Table 7.

Figure 16 shows the comparison of force–displacement graphs according to fracture toughness values from crash simulation. In the case of the cohesive zone model, it was confirmed that the structural rigidity of the aluminum/CFRP component decreased as the adhesive with a low fracture toughness value initially de-bonded (t = 0.003). In Case II, the crushing depth was reduced due to a high fracture toughness value; however, the results of the tiebreak contact condition show that Case I and Case II are quite similar. In Figure 17, the CFRP was not damaged, but the adhesive at the aluminum–CFRP interface was broken and separated due to low fracture toughness. In particular, as seen in the results of the cohesive zone model, it was determined that the debonding occurred in the center part of the aluminum–CFRP interface; however, as a result of the tiebreak contact condition, the CFRP was separated on one side of the joint part. In Case II (Figure 18), the fracture area of the adhesive was significantly reduced compared with previous results. Because of the high fracture toughness of the adhesive, bending failure was also confirmed in the CFRP. As a result, when a dissimilar material part in a crash situation uses unidirectional CFRP with high stiffness as a reinforcing material, the structural rigidity of the part can be maintained by delaying the interfacial separation between the aluminum and CFRP by using an adhesive with a high fracture toughness value.

## 5. Conclusions

Special numerical methods are required to represent the adhesively bonded joint parts within a crash simulation. This paper presents the finite-element analysis results of aluminum/CFRP hybrid joint parts using LS-DYNA. This work focused on capturing the failure behavior of the structural adhesive interface as well as the aluminum and composite laminates. To apply adhesive models with reliable crash analysis of aluminum/CFRP hybrid adhesive joint components, fracture toughness tests were performed. The results of the finite-element analysis were compared to verify the validity of the structural adhesive modeling techniques. In addition, there was no decrease in the stiffness of the structure due to damage to the composite material. The results are summarized as follows: (1)A test setup for investigating the response of aluminum/CFRP structure was proposed in crash situations. The failure behaviors of the aluminum and CFRP were observed at the corners of the aluminum and the center of the CFRP. From the graphs of force–displacement, it was confirmed that the load and the stiffness of the structure decreased slightly due to the failure of the CFRP as well as the debonding between the aluminum and the CFRP;(2)A finite-element analysis model was constructed by selecting a material model suitable for the material characteristics of the aluminum/CFRP joint parts. The material model MAT54 in LS-DYNA was employed to simulate the failure of CFRP in a practical design process since it requires simple input parameters. For aluminum, the commercialized material model MAT24 was used to reflect the elastic–plastic behavior. The fracture toughness tests were performed for material models of structural adhesive. The obtained results were values of 2.010 kJ/m^2^ for Mode I and 7.666 kJ/m^2^ for Mode II;(3)Modeling techniques for structural adhesives between different materials (aluminum and CFRP) were proposed. The two adhesive modeling techniques proposed are particularly well suited for numerical analyses of adhesive joints in large structures since they provide a compromise between accuracy and computational costs. A crash analysis was performed to verify the reliability of the structural adhesive modeling techniques. The results of the two types of adhesive modeling techniques were similar for crash simulation;(4)To study the effects of mesh sizes, several analyses were carried out for element sizes 1 mm, 2 mm, and 5 mm. A mesh size of ≤2 mm is necessary to obtain converged solutions. The simulation results of coarse mesh, sized 5 mm, significantly over-predicted the experimental results. In addition, it was not possible to observe a decrease in the stiffness of the aluminum/CFRP component because there was no failure of CFRP in the simulation results of coarse mesh sized 5 mm;(5)The results of the finite-element analysis were compared and analyzed to confirm the impact strength of the aluminum/CFRP hybrid adhesive joint parts according to the adhesive fracture toughness values and the effect on adhesive failure. The numerical analysis results showed that the adhesive plays a critical role in maintaining the structural stiffness in a crash situation of the component when composite materials with relative stiffness are used as reinforcement in dissimilar material parts, such as aluminum and CFRP.

## Figures and Tables

**Figure 1 polymers-13-03364-f001:**
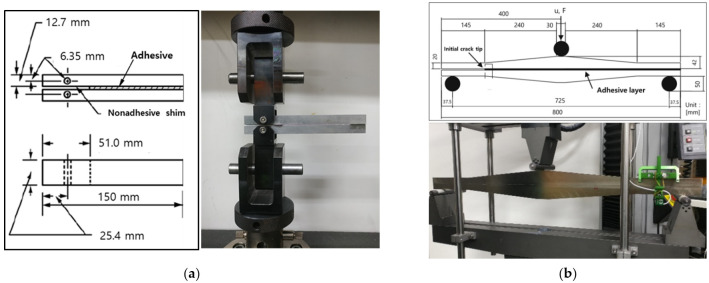
Dimensions of the specimen and fracture toughness test setup: (**a**) Mode I (**b**) Mode II.

**Figure 2 polymers-13-03364-f002:**
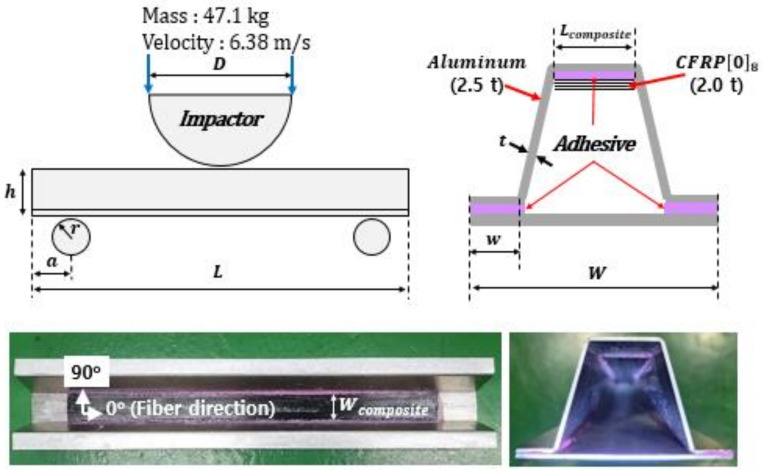
Configurations of aluminum/CFRP component test specimen.

**Figure 3 polymers-13-03364-f003:**
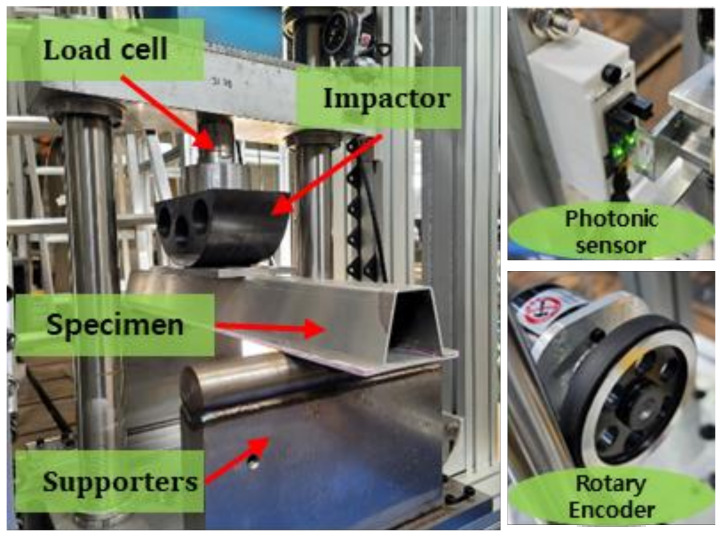
Experimental setup for aluminum/CFRP component test and sensors.

**Figure 4 polymers-13-03364-f004:**
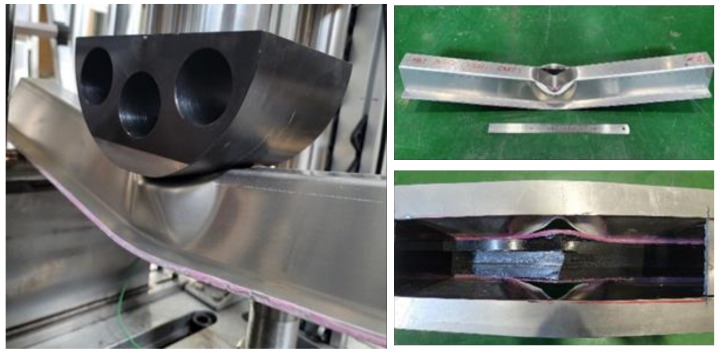
Failure of aluminum/CFRP component after the crash test.

**Figure 5 polymers-13-03364-f005:**
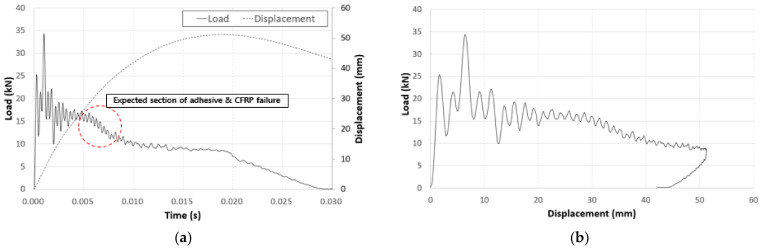
Graphs of the component crash test: (**a**) force–time curve and (**b**) force–displacement curve.

**Figure 6 polymers-13-03364-f006:**
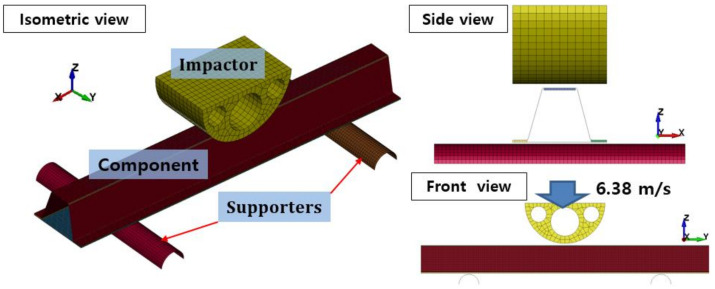
Finite-element model configuration of the component test.

**Figure 7 polymers-13-03364-f007:**
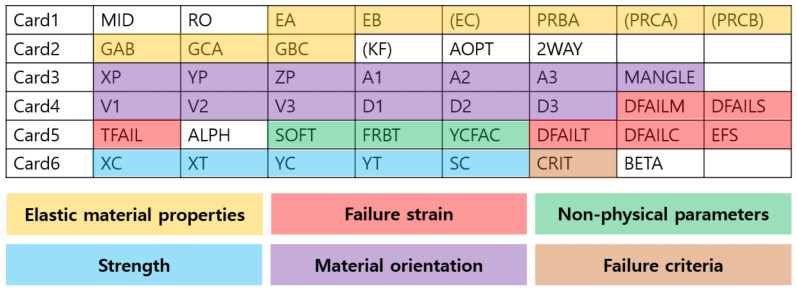
Overview of input parameters for the CFRP (MAT54).

**Figure 8 polymers-13-03364-f008:**
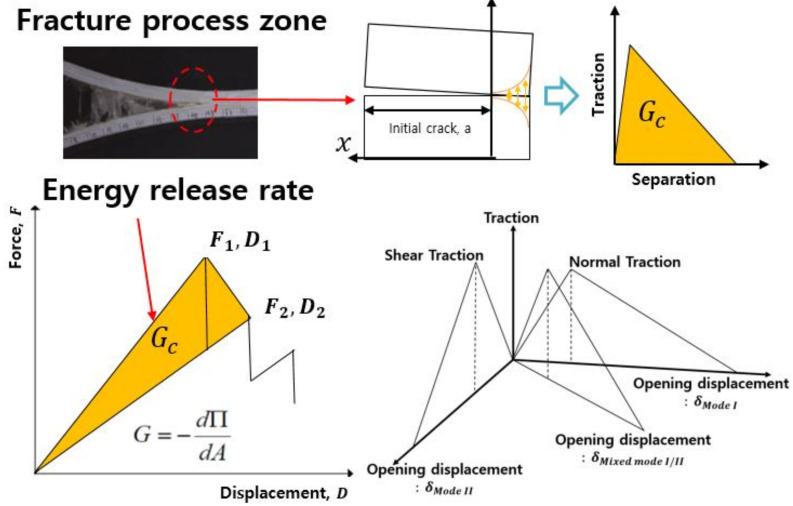
Cohesive law, energy release rate, and bilinear traction–separation law for material card MAT138.

**Figure 9 polymers-13-03364-f009:**
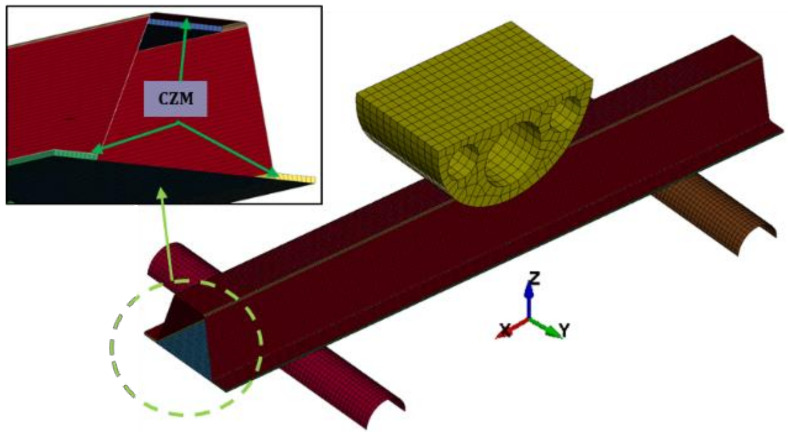
Finite-element cohesive zone model.

**Figure 10 polymers-13-03364-f010:**
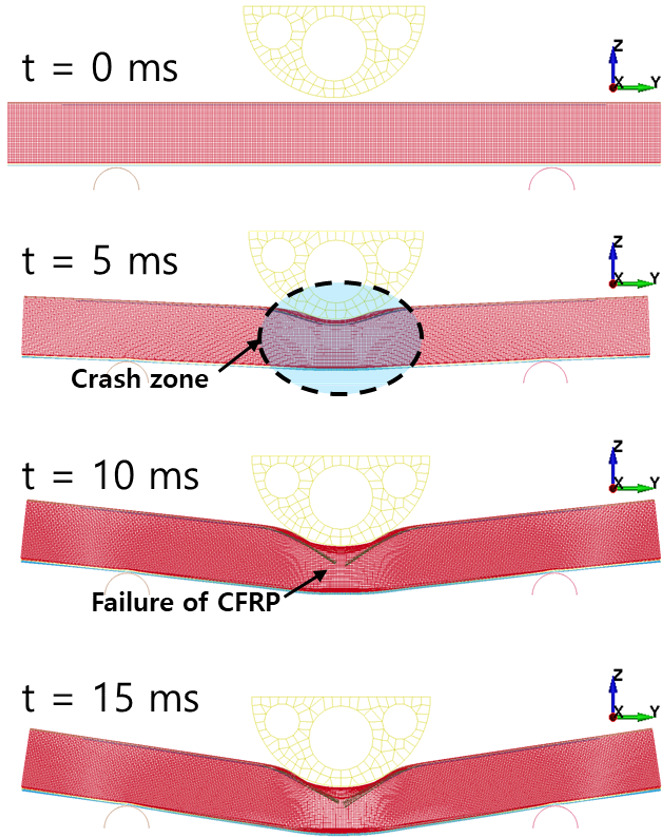
Sequential images of crash simulation.

**Figure 11 polymers-13-03364-f011:**
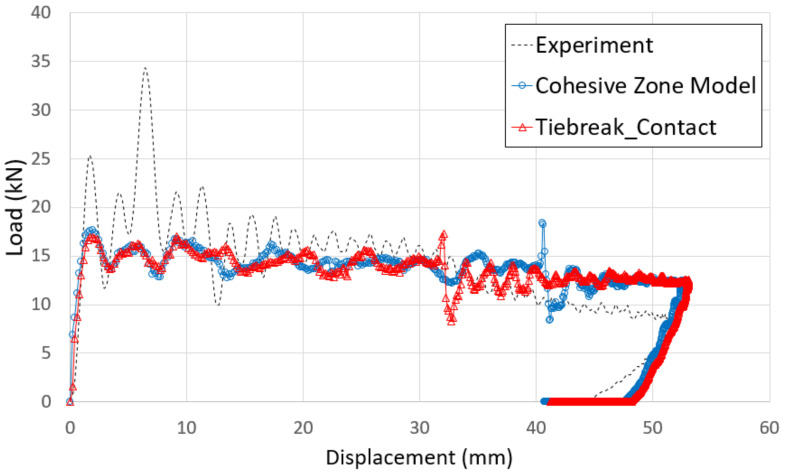
Comparison of force–displacement curves from the experiment and crash simulation.

**Figure 12 polymers-13-03364-f012:**
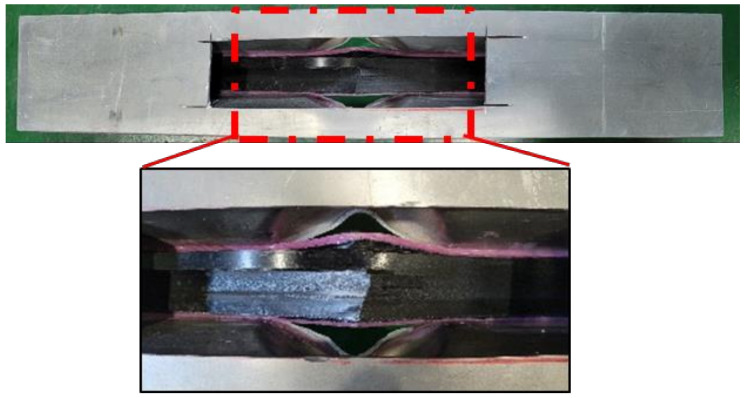
Failure of composite laminates from the experiment.

**Figure 13 polymers-13-03364-f013:**
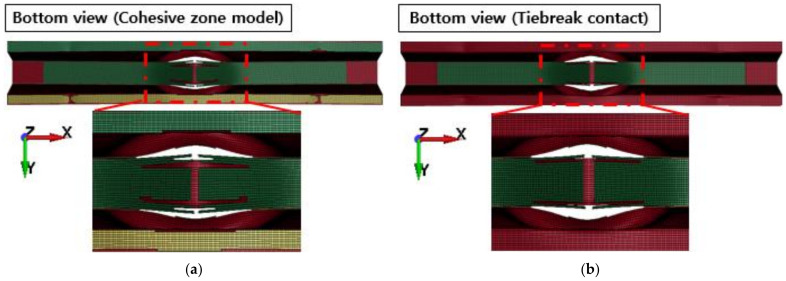
Failure of composite laminates: (**a**) cohesive zone model and (**b**) tiebreak contact condition.

**Figure 14 polymers-13-03364-f014:**
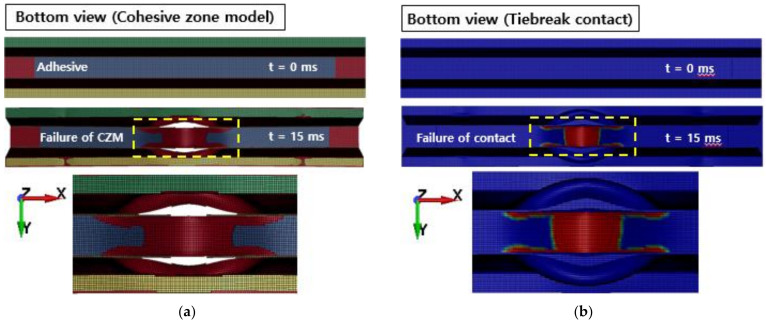
Failure of adhesive: (**a**) cohesive zone model and (**b**) tiebreak contact condition.

**Figure 15 polymers-13-03364-f015:**
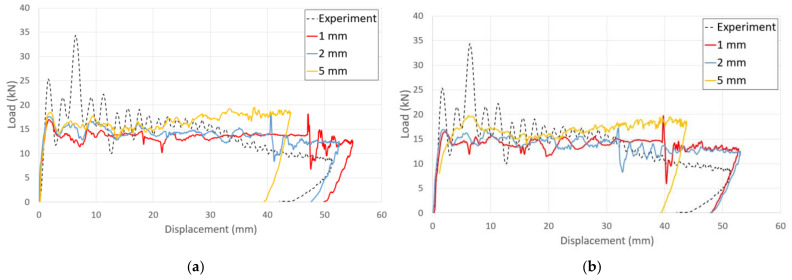
Force–displacement curves obtained for simulation with different mesh sizes: (**a**) cohesive zone model (**b**) tiebreak contact condition.

**Figure 16 polymers-13-03364-f016:**
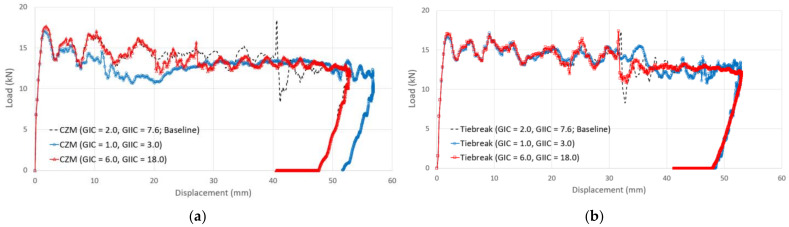
Comparison of force–displacement curves according to fracture toughness values from the crash simulation: (**a**) cohesive zone model (**b**) tiebreak contact condition.

**Figure 17 polymers-13-03364-f017:**
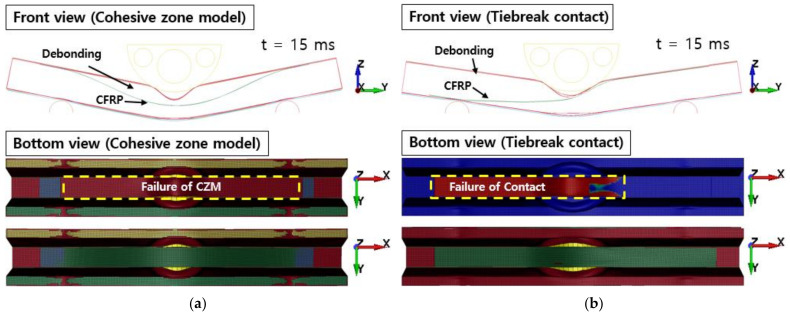
Failure of adhesive in Case I: (**a**) cohesive zone model (**b**) tiebreak contact condition.

**Figure 18 polymers-13-03364-f018:**
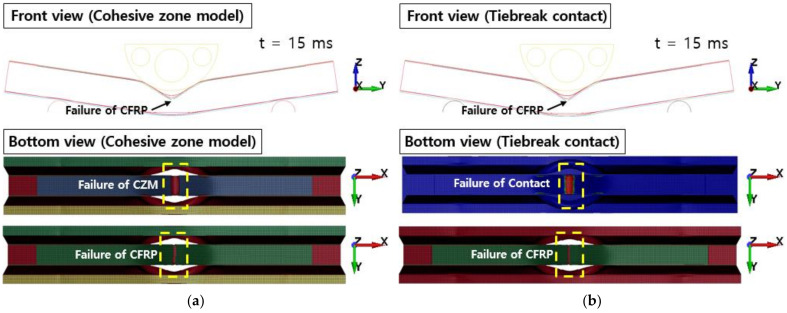
Failure of adhesive and composite laminates in Case II: (**a**) cohesive zone model (**b**) tiebreak contact condition.

**Table 1 polymers-13-03364-t001:** Mechanical properties of aluminum 5052-O [18,19,20].

	Modulus (GPa)	Tangent Modulus ^1^ (GPa)	Yield Strength (MPa)	Poisson’s Ratio
Aluminum 5052-O	70.3	2.2	89.6	0.33

^1^ Tangent modulus: Young’s modulus of plastic deformations.

**Table 2 polymers-13-03364-t002:** Mechanical properties of CFRP.

	Tensile Test		Compression Test		Shear Test
	Longitudinal	Transverse	Longitudinal	Transverse	
Modulus (GPa)	171	9.6	164	11.2	6.9
Strength (MPa)	2886	29	1367	195	73.6
Poisson’s ratio	0.28	-	-	-	-

**Table 3 polymers-13-03364-t003:** Fracture toughness of structural adhesive.

	Modulus (GPa)	Strength (MPa)	Energy Release Rate (kJ/m^2^)
Mode I	2.2	33.9	2.010 ^2^
Mode II	2.2	35.0	7.666 ^2^

^2^ Single measurement point.

**Table 4 polymers-13-03364-t004:** Dimensions of aluminum/CFRP component test specimen.

Dimension	Length (mm)	Description
*D*	200	Diameter of impactor
*L*	720	Length of beam
*L_composite_*	600	Length of composite plate
*r*	25	Radius of supports
*a*	120	Placement of supports
*h*	71.8	Height of beam
*W*	120	Width of beam
*w*	22.8	Width of beam
*W_composite_*	40	Width of composite plate
*t*	2.5	Thickness of aluminum 5052-O

**Table 5 polymers-13-03364-t005:** Energy absorption and crushing depth for simulation.

	Energy Absorption (J)	Crushing Depth ^3^ (mm)
Experimental	729	51.3
Cohesive zone model	704	52.5
Tiebreak contact condition	701	53.1

^3^ Maximum displacement of the impactor.

**Table 6 polymers-13-03364-t006:** Energy absorption and crushing depth for simulation with different mesh sizes.

	Mesh Size (mm)	Energy Absorption (J)	Crushing Depth ^4^ (mm)
Cohesive zone model	1	703	54.9
	2	704	52.5
	5	693	44.0
Tiebreak contact condition	1	698	52.7
	2	701	53.1
	5	694	43.8

^4^ Maximum displacement of the impactor.

**Table 7 polymers-13-03364-t007:** Input parameters of fracture toughness values for a parametric study.

	Modeling Method	
	Cohesive Zone Model	Contact Tiebreak
Case I	G_IC_ = 1.0 kJ/m^2^ G_IIC_ = 3.0 kJ/m^2^	
Case II	G_IC_ = 6.0 kJ/m^2^ G_IIC_ = 18.0 kJ/m^2^	

## Data Availability

The data presented in this study are available upon request from the corresponding author.

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
