# Peer review of "Crash Analysis of Aluminum/CFRP Hybrid Adhesive Joint Parts Using Adhesive Modeling Technique Based on the Fracture Mechanics"

_polymers, 2021, doi:10.3390/polym13193364_

Round 1

Reviewer 1 Report

In general, the article is well-organized and exhibits good quality. The list of detailed remarks is given below:

  1. The introduction part is a little bit poor. Please make it wider (there is many references about adhesive joints of aluminum-based materials).
  2. Table 2 - please add references (or marked, that obtained values are results of own investigations).
  3. Table 3 - average values or single measurement point? Please comment it.
  4. Fig. 3 - in the description of "t" please change, that it is not steel material.
  5. Fig. 12 and Table 5 - these values are different from an experiment values, please comment it.

Reviewer 2 Report

The manuscript is interesting. My comments:

  1. In introduction, please include a short literature review of similar studies and clear which is the innovation of your research.
  2. Conclusions need improvement. Please add useful conclusions.
  3. In text, several typos appear, please correct.
  4. Fig.8 is of low quality, please improve.
  5. Delete table 8 and add the information in text.

Reviewer 3 Report

This is interesting research. However, the manuscript needs some modifications as follows:

  1. Figure 3 should be improved to be accurate and readable.
  2. Verification of Finite-element simulation should be extended.
  3. The accuracy of the finite element results is affected by the mesh size. The analysis of mesh sensitivity should be added. The following references should be discussed in Introduction and Section 4.4:
    a) Reviews on Advanced Materials Science, Volume 60, Issue 1, Pages 615 - 6301 January 2021.
    b) Mechanics of Advanced Materials and Structures, In Press, Published online: 26 Aug 2021. DOI: 10.1080/15376494.2021.1952663
    c) Thin-Walled Structures 158 (2021) 107030.
  4. The difference in the force-displacement curves from the experiment and crash simulation (Fig. 12) should be improved by improving the mesh size and boundary conditions.
  5. Section 4.5 should be extended.

Round 2

Reviewer 2 Report

good, accept

Reviewer 3 Report

The authors have successfully addressed all my comments.  Therefore, I recommend the publication of this manuscript.